# CLIP-to-Seg Distillation for Inductive Zero-Shot Semantic Segmentation

## Abstract

CLIP has greatly advanced zero-shot segmentation by leveraging its strong visual-language association and generalization capability. However, directly adapting CLIP for segmentation often yields suboptimal results due to inconsistencies between image and pixel-level prediction objectives. Additionally, merely combining segmentation and CLIP models often leads to disjoint optimization, introducing significant computational overhead and additional parameters. To address these issues, we propose a novel CLIP-to-Seg Distillation approach, incorporating global and local distillation to flexibly transfer CLIP's powerful zero-shot generalization capability to existing closed-set segmentation models. Global distillation leverages CLIP's CLS token to condense segmentation features and distills high-level concepts to the segmentation model via image-level prototypes. Local distillation adapts CLIP's local semantic transferability to dense prediction tasks using object-level features, aided by pseudo-mask generation for latent unseen class mining. To further generalize the CLIP-distilled segmentation model, we generate latent embeddings for the mined latent classes by coordinating their semantic embeddings and dense features. Our method equips existing closed-set segmentation models with strong generalization capabilities for open concepts through effective and flexible CLIP-to-Seg distillation. Without relying on the CLIP model or adding extra computational overhead/parameters during inference, our method can be seamlessly integrated into existing segmentation models and achieves state-of-the-art performance on multiple zero-shot segmentation benchmarks.

## 1 Introduction

In recent years, semantic segmentation has advanced rapidly, benefiting from deep learning technologies (Long et al., 2015; Chen et al., 2018). However, conventional semantic segmentation models are heavily data-dependent, requiring large volumes of annotated images to achieve satisfactory performance. Collecting these images and annotations is both time-consuming and expensive.

To address this challenge, zero-shot semantic segmentation has been proposed and has gained significant attention (Xian et al., 2019; Gu et al., 2020). In zero-shot semantic segmentation, models are trained on seen classes and must generalize to unseen classes during inference, relying solely on their text descriptions. To accomplish this, existing methods (Ding et al., 2022a; Zhou et al., 2023) typically utilize Vision-Language models with strong zero-shot generalization capabilities, such as CLIP (Radford et al., 2021), for pixel-level segmentation tasks.

To effectively adapt CLIP for segmentation, existing methods can be categorized into two groups: one-stage methods and two-stage methods, as shown at the top of Fig. 1. In one-stage methods (Han et al., 2023a; Zhou et al., 2023), to maintain CLIP's generalization capability, the adaptation module or trainable prompts are often inserted after the frozen CLIP visual encoder to adapt the dense tokens for segmentation. Two-stage methods (Ding et al., 2022a; Xu et al., 2022) typically require a pre-trained, class-agnostic object proposer to identify latent objects in an image. These object proposals are then fed into the frozen CLIP visual encoder for classification generalization.

Despite their effectiveness, both approaches exhibit inherent limitations. In one-stage methods, CLIP is primarily optimized for capturing global context through the CLS token, but it lacks the spatial information required to capture fine-grained local details necessary for precise segmentation.

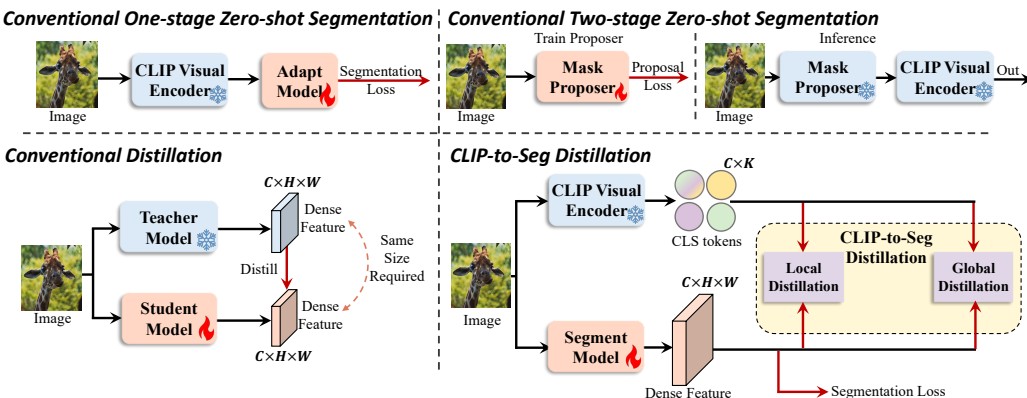

Figure 1: Comparisons between CLIP-to-Seg distillation and other methods. **Top Left**: Conventional one-stage zero-shot segmentation, which directly adapts CLIP for dense prediction tasks. **Top Right**: Conventional two-stage zero-shot segmentation, where a proposer is trained and frozen CLIP is used to classify the proposals. **Bottom Left**: Conventional knowledge distillation methods require the student and teacher models to be the same size. **Bottom Right**: Our CLIP-to-Seg distillation transfers CLIP's knowledge to segmentation models and does not rely on CLIP during inference, resulting in high inference performance and efficiency.

However, dense prediction tasks prioritize high-quality pixel-level parsing over image-level understanding, creating a mismatch between task requirements and CLIP's capabilities, thus limiting the effectiveness of one-stage methods. Two-stage methods primarily suffer from the disjointed optimization between mask proposal generation and CLIP's class recognition. Additionally, two-stage methods are computationally expensive, as they require processing both mask proposal generation and per-proposal classification.

To address the limitations of both approaches, we propose a framework that **1)** achieves high-quality segmentation without incurring additional computational costs during inference, and **2)** simultaneously maintains strong zero-shot generalization capabilities for open concepts. We begin by revisiting closed-set segmentation models, which are highly optimized for capturing local details essential for precise segmentation (Xie et al., 2021; Guo et al., 2022). However, these models tend to overfit to seen classes due to the absence of data for unseen classes, despite their effectiveness at segmenting objects. Recent methods have attempted to overcome this limitation by employing knowledge distillation to transfer CLIP's zero-shot capabilities to task-specific models for adapting various downstream tasks (Huang et al., 2024; Han et al., 2023b). However, these approaches require matching feature sizes between teacher and student models (see bottom left of Fig. 1), making it difficult to transfer CLIP's knowledge from a few tokens to the dense features of various segmentation models.

These limitations motivate us to propose CLIP-to-Seg (C2S) distillation, which integrates global, local, and latent embedding distillation to transfer CLIP's vision-language matching capabilities to pixel-level segmentation models. Global distillation adaptively aggregates dense features into image-level prototypes based on their similarity to global CLS tokens, and then performs efficient prototype-token distillation to transfer CLIP's zero-shot generalization capabilities to the segmentation model. However, this image-level distillation may overlook non-dominant classes and fine-grained object details. To address this, we propose local distillation to adapt CLIP's local semantic transferability to dense prediction tasks through object-level prototypes. CLIP's local tokens and the segmentation model's local prototypes are generated by mining latent unseen classes, aided by pseudo mask generation. To further generalize the CLIP-distilled segmentation model for unseen classes, we generate latent embeddings for the mined latent classes to help the model perceive their real data statistics during training. The latent embedding generation coordinate the semantic embeddings and dense features of the mined latent classes, distilling suitable prototypes for subsequent mask prediction and generalization on unseen classes.

Unlike existing approaches that adapt the CLIP visual encoder (Zhou et al., 2022; 2023) or ensemble with CLIP during inference (Ding et al., 2022a; Han et al., 2023a), our method can be seamlessly integrated into existing closed-set segmentation models without relying on the CLIP model or introducing additional computational overhead/parameters at inference. Our method achieves state-of-

the-art performance on multiple zero-shot segmentation benchmarks when incorporated with powerful segmentation models such as Segformer (Xie et al., 2021) and SegNeXt (Guo et al., 2022). In summary, our contributions are:

– We propose a novel CLIP-to-Seg distillation method to effectively and efficiently adapt CLIP for segmentation by integrating global and local distillation collaboratively.

– We propose a novel latent embedding generation method to further help the CLIP-distilled segmentation model to generalize well on latent unseen classes.

– Without introducing additional parameters or computational overhead during inference, our method can be flexibly integrated into current powerful segmentation models and achieves state-of-the-art performance on multiple zero-shot segmentation benchmarks.

## 2 RELATED WORKS

**Close-set Semantic Segmentation.** Close-set segmentation assumes fully annotated images and focuses on the performance of predefined categories within a specific dataset. Existing methods are typically divided into pixel-level classification and mask-level classification. In pixel-level classification, FCN (Long et al., 2015), the first fully convolutional network for end-to-end semantic segmentation, established the paradigm for pixel-level methods. Since FCN, many works, *e.g.*, DeepLab series (Chen et al., 2018; 2017), Deformable convolution (Dai et al., 2017), aim to enlarge the receptive field to further improve the performance of pixel-level methods. With the introduction of self-attention (Vaswani et al., 2017) and ViT (Dosovitskiy et al., 2020), many approaches (Zheng et al., 2021; Xie et al., 2021; Guo et al., 2022; Liu et al., 2021) replaced the conventional convolutional backbone with self-attention-based models, achieving remarkable performance. An alternative approach treats semantic segmentation as a mask classification task. MaskFormer (Cheng et al., 2021a) and Mask2Former (Cheng et al., 2022) are two notable examples of this approach. Specifically, these models first generate object queries corresponding to potential objects. These object queries are then decoupled to perform classification and mask prediction tasks separately. Our method is applied to the more challenging task of zero-shot segmentation, which requires fewer annotations than close-set segmentation.

**Knowledge Distillation.** Knowledge distillation aims to transfer the capability of a larger teacher model to a student model for comparable performance to the teacher model with a smaller model size (Wang & Yoon, 2021). Existing methods are categorized into logits-based (Hinton, 2015; Yang et al., 2023; Touvron et al., 2021), feature-based (Huang et al., 2024; Tian et al., 2019; Quan et al., 2023), and relation-based approaches (He et al., 2023; Han et al., 2023b). With the rapid development of vision-language models (Radford et al., 2021; Jia et al., 2021; Zhang et al., 2023), certain methods aim to distill vision-language matching capabilities into other models (Huang et al., 2024; Quan et al., 2023; Pei et al., 2023). However, these methods distill knowledge between classification models, transferring it from one global context to another. Our method distills the knowledge from a classification model to a segmentation model where the knowledge is transferred from a global context to a local context across different feature sizes.

**Zero-shot Semantic Segmentation.** Since close-set segmentation requires pixel-level annotations, research focusing on reducing label dependency has gained significant attention. Before the VLMs, *e.g.*, CLIP (Radford et al., 2021), several works tried to bridge the gap between vision and language by projecting the features from vision models to the semantic space which is spanned by the large scale of texts (Gu et al., 2020; Xian et al., 2019). The emergence of large-scale VLMs, such as CLIP (Radford et al., 2021) and ALIGN (Jia et al., 2021), has revolutionized zero-shot tasks. Due to their impressive zero-shot ability, researchers aim to transfer this ability to downstream tasks. Leveraging visual prompt tuning (Jia et al., 2022) and adapters (Houlsby et al., 2019), existing methods are categorized into one-stage and two-stage approaches. One-stage methods introduce trainable parameters or modules to adapt VLMs for semantic segmentation (Xu et al., 2023b; Li et al., 2022; Ghiasi et al., 2022; Zhou et al., 2023; 2022; Guo et al., 2023; Ding et al., 2022a;b; Qin et al., 2023; Yu et al., 2023; Wu et al., 2024). Two-stage methods train a mask-proposer (Cheng et al., 2022; 2021a) to find the potential objects in an image and utilize the proposed objects to finetune the VLMs or directly classify the objects (Xu et al., 2022; Shin et al., 2023; Zhou et al., 2022; Jiao et al., 2023; Xu et al., 2023a).

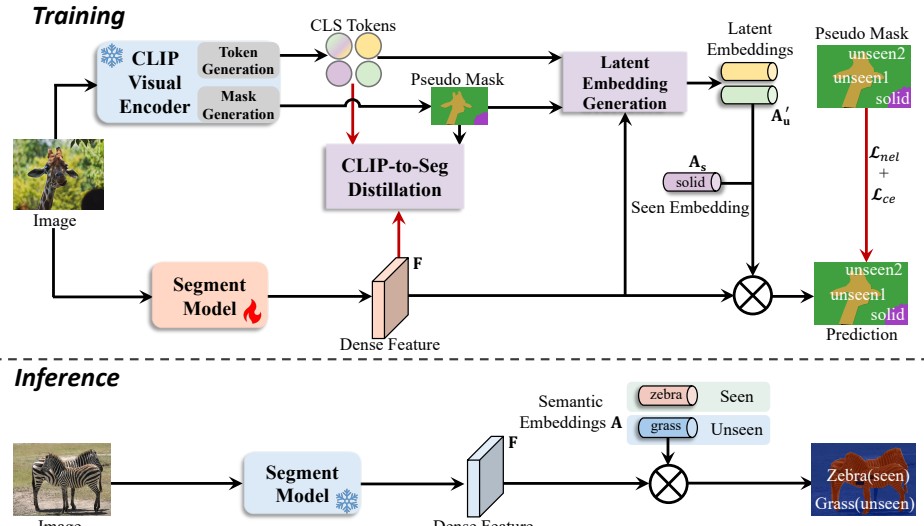

Figure 2: Overview of the CLIP-to-Seg distillation framework. First, the input image is passed through a frozen CLIP visual encoder to obtain both global and local CLS tokens, as well as pseudo masks for latent classes. The same image is then passed into a trainable segmentation model to extract dense features. CLIP's vision-language matching capabilities are transferred through the proposed CLIP-to-Seg distillation. To provide additional supervision for latent classes, we propose a latent embedding generation method to synthesize semantic embeddings for latent classes. During inference, our method does not introduce any additional modules or parameters to the segmentation model and relies solely on the segmentation model, resulting in high inference efficiency.

Different from both types of CLIP-adapting paradigm that rely heavily on CLIP during inference, we propose a CLIP-to-Seg distillation method to transfer the vision-language capability to any pixel-level segmentation model, enabling them to employ zero-shot semantic segmentation without CLIP in inference. Although some methods distill the text relationships to the vision space (He et al., 2023; Han et al., 2023b), their methods works under a relaxed condition where all the text embeddings can be accessed. Meanwhile, some object detection methods also try to distill the knowledge from CLIP to detection models (Gu et al., 2022; Gao et al., 2022). However, their methods need to train an additional mask proposer and a detailed description of the input image.

# 3 METHODS

**Task Definition.** Before presenting our method, we first define the task of Zero-shot Semantic Segmentation (ZSS). Formally, let $\mathcal{D} = \left\{ \mathbf{X}^i, \mathbf{Y}^i \right\}_{i=1}^{M}$ represent a dataset, where $\mathbf{X}$ are the input images, $\mathbf{Y}$ are the corresponding pixel-level annotations, and $\mathbf{A} \in \mathcal{R}^{N \times D}$ is a set of semantic embeddings for all categories, with $N$ representing the total number of classes and $D$ the dimensionality of the embeddings. The semantic embeddings $\mathbf{A}$ are partitioned into two disjoint subsets: seen class embeddings $\mathbf{A}_s \in \mathcal{R}^{N_s \times D}$ and unseen class embeddings $\mathbf{A}_u \in \mathcal{R}^{N_u \times D}$, where $\mathbf{A}_s \cap \mathbf{A}_u = \varnothing$ and $N_s + N_u = N$. Since seen and unseen classes frequently co-occur in images, removing those containing unseen categories is impractical for training. Therefore, in ZSS, only the annotations for unseen classes are removed. ZSS can be categorized into two settings based on the availability of unseen class embeddings $\mathbf{A}_u$: *Inductive ZSS*, where unseen class embeddings are unavailable during training, and *Transductive ZSS*, where unseen class embeddings are accessible. In both settings, model performance is jointly evaluated on both seen and unseen categories during inference. In this work, we adopt the inductive ZSS setting, which is more challenging and closer to real-world applications.

**Method Overview.** The overview of methods is shown in Fig. 2. First, the input image is passed through a frozen CLIP visual encoder to extract CLS tokens and pseudo masks for seen and latent classes. Simultaneously, the same image is fed into a trainable segmentation model to extract dense features. Then, we apply the proposed CLIP-to-Seg (C2S) distillation to transfer CLIP's knowledge

to the segmentation model as illustrated in Sec. 3.2. Relying solely on C2S distillation may lead to suboptimal performance for the segmentation model. To address this, as described in Sec. 3.3, we propose a latent embedding generation method to synthesize semantic embeddings for latent classes. These synthetic embeddings help differentiate latent classes from other categories, providing pixel-level supervision for unannotated regions.

## 3.1 TOKEN GENERATION AND MASK GENERATION

The core idea of CLIP-to-Seg (C2S) distillation is to transfer CLIP's powerful vision-language alignment capabilities to a segmentation model regardless of the size differences between the CLIP and the segmentation model. To achieve this, we first generate CLS tokens, both global and local, which act as the teacher features during the distillation process, as shown in the top left of Fig. 3.

Given an input image $\mathbf{X}^{H \times W \times 3}$, we first pass it through the CLIP visual encoder to obtain the global CLS token $\mathbf{C}_g$. However, because CLIP inherently focuses on the global context, it may overlook less prominent classes within the image. To address this limitation and capture the semantics of all classes within an image, we additionally extract local CLS tokens. Specifically, for an image $\mathbf{X}$ with its corresponding pixel-level annotation $\mathbf{Y}$, we assume that annotations are available for all classes, including unseen ones. We first separate $\mathbf{Y}$ into non-overlapping class-specific masks based on unique categories, where $\mathbf{Y} = \{\mathbf{Y}_i\} i = 0^O$, with $\mathbf{Y}_i$ representing the binary mask for the $i$th class, and $O$ representing the number of unique classes in the image. Using these masks, we pool the original image $\mathbf{X}$ into class-specific sub-images. Each class-specific sub-image is then passed through the CLIP visual encoder to extract the corresponding local CLS tokens $\mathbf{C}_l$.

In practice, annotations for unseen classes are inaccessible, resulting in large unannotated areas within an image. We refer to the classes in these areas as latent classes, as they may either belong to unseen categories or are simply unannotated in the dataset. To further leverage the dense features of these latent classes, we propose a latent class mining algorithm that clusters the dense visual tokens from the CLIP visual encoder. Specifically, we first initialize seeds $\mathbf{S}$ by applying sliding windows of various sizes to average the dense tokens:

$$\mathbf{S} = \left\{ \sum_{u=i}^{i+o-1} \sum_{v=j}^{j+o-1} \frac{\mathbf{C}_d[u,v]}{o^2} \Big| o \in \mathcal{O}, \text{if } y[u,v] \in A_s \text{ then } \mathbf{C}_d[u,v] = 0 \right\}, \quad (1)$$

where $\mathbf{C}_d$ represents the CLIP visual dense tokens, and $i \in \{0, [o/2], [o], ..., [H_d - o]\}$ and $j \in \{0, [o/2], [o], ..., [W_d - o]\}$ denote the stride of the sliding windows. Here, $H_d$ and $W_d$ represent the size of $\mathbf{C}_d$, and $[\cdot]$ denotes the rounding operation. $\mathcal{O}$ denotes the set of window sizes. Based on these seeds, we apply K-Means clustering to the unannotated regions of $\mathbf{C}_d$ and merge clusters according to the cosine similarities between the updated seeds. The pseudo-code and merging details are provided in the ***Supplementary Materials***.

Once the latent classes are identified, we combine the given seen labels with the masks for the latent classes to create the pseudo masks $\mathbf{Y}_p$. Consequently, the local CLS tokens for latent classes can also be extracted.

## 3.2 CLIP-TO-SEG DISTILLATION

Recent methods have attempted to transfer CLIP's vision-language matching capabilities to other models using knowledge distillation (Huang et al., 2024; Han et al., 2023b). However, conventional knowledge distillation faces the challenge of requiring feature size matching between teacher and student models, which hinders knowledge transfer from CLIP to segmentation models. To overcome this limitation, we propose CLIP-to-Seg distillation, consisting of two components: global distillation and local distillation. We first introduce global distillation, which transfers CLIP's knowledge by aligning global CLS tokens with global feature prototypes. Specifically, as illustrated in the top right of Fig. 3, the input image is passed through a trainable segmentation model to extract dense features $\mathbf{F}^{D \times H \times W}$, where $D$ is the number of channels, and $H$ and $W$ are the height and width of the feature map, respectively. To compute the global prototype, $\mathbf{F}$ is reshaped to $D \times L$, where $L = H \times W$. The similarity $\mathbf{W}$ between $\mathbf{F}$ and the global CLS token $\mathbf{C}_g$ is computed as $\mathbf{W} = \text{Softmax}(\frac{\mathbf{C}_g^\top \mathbf{F}}{\sqrt{D}})$, where $\mathbf{W}^{1 \times L} \in [0, 1]$, and the softmax is applied along the second dimension

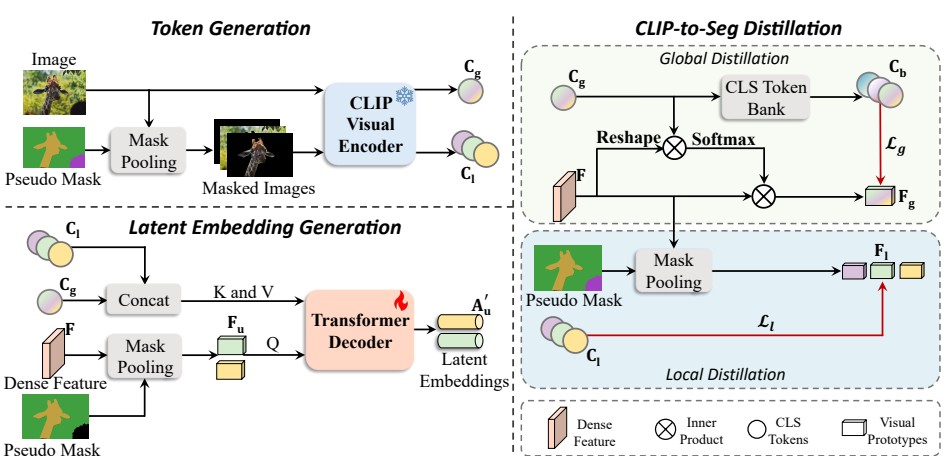

Figure 3: The process of token generation, latent embedding generation and CLIP-to-Seg ditillation.

of $\mathbf{W}$. This similarity is then used to weigh the contributions of each dense feature in generating the global feature prototype $\mathbf{F}_g$, where $\mathbf{F}_g = \mathbf{W}\mathbf{F}^\top$ .

Inspired by the memory buffer mechanism in contrastive learning to provide more negative pairs (Wu et al., 2018), we also introduce a CLS token bank to store CLS tokens from previous iterations. Let $\mathcal{V} = \{\mathbf{C}_i\}$ denote the CLS token bank. In each iteration, before updating the model parameters, we enqueue the current CLS token $\mathbf{C}$ into $\mathcal{V}$ and dequeue the oldest CLS tokens. Finally, we align the global prototype $\mathbf{F}_g$ with the CLS token bank $\mathbf{C}_b$ using InfoNCE (Oord et al., 2018),

$$\mathcal{L}_g = \sum_{i=0}^{M+1} \frac{\exp(\mathbf{F}_g^\top \mathbf{c}_i / \tau)}{\sum_{j=0}^{M+1} \exp(\mathbf{F}_g^\top \mathbf{c}_j)/\tau)}, \tag{2}$$

where $\mathbf{c}_j \in \mathbf{C}_b$, and $\tau$ denotes the temperature used for contrastive loss. However, due to CLIP's focus on the global context, it may overlook less prominent classes, failing to transfer accurate semantics to the dense features associated with them. To remedy this, we propose the local distillation methods. as shown in the bottom left Fig. 3.

Local distillation seeks to transfer semantics overlooked by the global CLS tokens to their corresponding dense features by aligning local feature prototypes with the local CLS tokens. Specifically, given the pseudo mask $\mathbf{Y}_p$, we first pool the dense features from these areas and average the class-specific features to obtain the local prototypes $\mathbf{F}_l$:

$$\mathbf{F}_l = \left\{ f_l = \frac{\sum_{H,W} \mathbf{F}[\mathbb{1}(y_i = l)]}{\sum_{H,W}[\mathbb{1}(y_i = l)]} \middle| y_i \in \mathbf{Y}_p \right\}, \tag{3}$$

where $\mathbb{1}(y_i = l)$ is an indicator function that selects pixels belonging to class $l$. Finally, given $\mathbf{C}_l$, we apply InfoNCE (Oord et al., 2018) to align the local prototypes $\mathbf{F}_l$ with the local CLS tokens $\mathbf{C}_l$,

$$\mathcal{L}_l = \sum_{i=0}^{P} \frac{\exp(\mathbf{f}_i^\top \mathbf{c}_i / \tau)}{\sum_{j=0}^{P} \exp(\mathbf{f}_i^\top \mathbf{c}_j)/\tau)}, \tag{4}$$

where $\mathbf{f} \in \mathbf{F}l$ and $\mathbf{c} \in \mathbf{C}l$, with the positive pairs being the local prototypes and CLS tokens from the same class in $\mathbf{Y}_p$. By transferring CLIP's knowledge to segmentation models through C2S distillation, the model's generalization is further improved, reducing overfitting to seen classes.

### 3.3 LATENT EMBEDDING GENERATION

Although CLIP's vision-language matching capabilities are effectively transferred to segmentation models, the inaccessibility of unseen semantic embeddings leaves large portions of dense features without pixel-level supervision, resulting in suboptimal optimization of the segmentation model. To address this, we propose Latent Embedding Generation, which generates synthetic semantic embeddings for latent classes by calibrating the local feature prototypes with their corresponding local CLS tokens, as shown in the bottom left of Fig. 3.

Given $\mathbf{Y}_p$, we first select the binary masks $\mathbf{Y}_u$ corresponding to the latent classes. Next, we use Eq. 3 to replace $\mathbf{Y}_p$ with $\mathbf{Y}_u$ to generate visual prototypes for the latent classes. We then feed $\mathbf{F}_u$ into a transformer decoder as Query and input the global and local CLS tokens as Key and Value to generate the latent prototypes $\mathbf{A}'_u$. The latent prototypes $\mathbf{A}'_u$ are treated equivalently to seen semantic embeddings and are used to distinguish between the seen and latent classes. Formally, the class scores for seen and latent categories are $\mathbf{X}s = \alpha \cdot \mathbf{F}^\top \mathbf{A}s$ and $\mathbf{X}u = \beta \cdot \cos(\mathbf{F}, \mathbf{A}u')$, where $\alpha$ and $\beta$ are hyperparameters that control the scale of unseen categories. Note that, since the pseudo labels and potential prototypes are not entirely precise, cosine similarity helps prevent overemphasis on misclassification and aids in distinguishing between seen and potential categories. We then concatenate the logits for both seen and unseen classes as $\mathbf{X}logits = \text{cat}(\mathbf{X}s, \mathbf{X}_u)$, where 'cat' denotes concatenation along the class dimension. Finally, $\mathbf{Y}_p$ is used to provide pixel-level supervision to the dense features through:

$$\mathcal{L}_p = \mathcal{L}_{nel}(\mathbf{X}_{logits}, \mathbf{Y}_f) + \mathcal{L}_{ce}(\mathbf{X}_{logits}, \mathbf{Y}_f). \tag{5}$$

where $\mathcal{L}nel$ refers to the NEL loss (Zhou et al., 2023), and $\mathcal{L}ce$ denotes the cross-entropy loss.

### 3.4 Training Objective and Inference

**Training Objective.** To recap, the training objectives of CLIP-to-Seg distillation are:

$$\mathcal{L} = \mathcal{L}_g + \mathcal{L}_l + \mathcal{L}_p, \tag{6}$$

**Inference.** Since the vision-language matching capability has already been transferred from CLIP to the backbone during training, we do not need to rely on CLIP at inference time. The backbone, having learned to align dense features with semantic embeddings, can independently produce accurate segmentation results, including for unseen categories.

## 4 Experiments

**Dataset.** To evaluate the effectiveness of our method, we select three representative benchmarks: PASCAL VOC (Everingham et al., 2015), COCO-Stuff (Caesar et al., 2018), and PASCAL Context (Mottaghi et al., 2014) to conduct our experiments on zero-shot semantic segmentation (ZSS). The split of seen and unseen categories follows the setting of the previous works (Ding et al., 2022a; Zhou et al., 2023; 2022). *PASCAL VOC* consists of 10,582 images for training and 1,449 images for validation. Note that we convert the 'background' category to the 'ignored'. For this dataset, there are 15 seen categories and 5 unseen categories. *COCO-Stuff* contains 171 categories totally. As in previous settings, 171 categories are split into 156 seen and 15 unseen categories. Besides, for the training dataset, there are 118,287 images and 5,000 images for testing. *PASCAL Context* includes 4,996 images for training and 5,104 images for testing. For the zero-shot semantic segmentation task, the dataset is split into 49 seen categories and 10 unseen categories.

**Implementation Details.** The proposed methods are implemented on the MMsegmentation (Contributors, 2020). The CLIP model applied in our method is based on the ViT-B/16 model and the channel ($C$) of the output text features is 512. All the experiments are conducted on 8 V100 GPUs and the batch size ($B$) is set to 16 for all three datasets. For all these three datasets, the size of the input images is set as $512 \times 512$. The iterations are set to 20k, 40k, and 80k for PASCAL VOC, PASCAL Context, and COCO-Stuff respectively. The optimizer is set to AdamW with the default training schedule in the MMSeg toolbox (Contributors, 2020). In addition, the size of CLS tokens banks is set as 24, Other settings can be seen in ***Supplementary materials***.

To evaluate the performance of both seen and unseen categories, we apply the harmonic mean IoU (hIoU) following previous works (Zhou et al., 2023; Ding et al., 2022a; Bucher et al., 2019). The relationship between mIoU and hIoU is $hIoU = \frac{2 \cdot sIoU \cdot uIoU}{sIoU + uIoU}$ where $sIoU$ and $uIoU$ indicate the mIoU of the seen categories and unseen categories, respectively. Besides the hIoU, $sIoU$ and $uIoU$ are also applied. Frames Per Second (FPS) on one RTX 3090 is the metric for inference speed.

### 4.1 Comparison with Sota-of-the-arts

We apply our method with three representative closed-set segmentation models, *i.e.*, SegNext (Guo et al., 2022), SETR (Zheng et al., 2021) and Segformer (Xie et al., 2021) by distilling the knowledge of CLIP to these segmentation models. We compare the performance with the state-of-the-art

Table 1: Comparison with state-of-the-art methods where the **bold** and the underline indicates the best and the second-best performance.

| Models | Backbone | PASCAL VOC | | | COCO-Stuff | | | PASCAL Context | | |
|---|---|---|---|---|---|---|---|---|---|---|
| | | hIoU | sIoU | uIoU | hIoU | sIoU | uIoU | hIoU | sIoU | uIoU |
| SPNet (Xian et al., 2019) | | 26.1 | 78.0 | 15.6 | 14.0 | 35.2 | 8.7 | - | - | - |
| ZS3 (Bucher et al., 2019) | | 28.7 | 77.3 | 17.7 | 15.0 | 34.7 | 9.5 | 15.8 | 20.8 | 12.7 |
| CaGNet (Gu et al., 2020) | ResNet101 (He et al., 2016) | 39.7 | 78.4 | 26.6 | 18.2 | 33.5 | 12.2 | 21.2 | 24.1 | 18.5 |
| SIGN (Cheng et al., 2021b) | | 41.7 | 75.4 | 28.9 | 20.9 | 32.3 | 15.5 | - | - | - |
| Joint (Baek et al., 2021) | | 45.9 | 77.7 | 32.5 | - | - | - | 20.5 | 33.0 | 14.9 |
| ZegFormer (Ding et al., 2022a) | | 73.3 | 86.4 | 63.6 | 34.8 | 36.6 | 33.2 | - | - | - |
| Zzseg (Xu et al., 2022) | | 77.5 | 83.5 | 72.5 | 37.8 | 39.3 | 36.3 | - | - | - |
| ZegCLIP (Zhou et al., 2023) | | 84.3 | 91.9 | 77.8 | 40.8 | 40.2 | 41.4 | 49.9 | 46.0 | 54.6 |
| DeOP (Han et al., 2023a) | ViT-B (Dosovitskiy et al., 2020) | 80.8 | 88.2 | 74.6 | 38.2 | 38.0 | 38.4 | - | - | - |
| OTSeg+ (Ye et al., 2024) | | 87.1 | **93.3** | 81.6 | 41.5 | 41.3 | 41.8 | 57.7 | **55.2** | 60.4 |
| CLIP-RC (Zhang et al., 2024) | | 88.4 | 92.8 | 84.4 | 41.2 | 40.9 | 41.6 | 51.9 | 47.5 | 51.9 |
| | SegNeXt-B (Guo et al., 2022) | 89.3 | 91.2 | 87.4 | 42.5 | 43.1 | 41.9 | 57.6 | 53.3 | **62.8** |
| Ours | Setr-B (Zheng et al., 2021) | **90.7** | 92.3 | **89.2** | **44.8** | **43.8** | **45.9** | 56.3 | 52.4 | 60.8 |
| | Segformer-B4 (Xie et al., 2021) | 88.7 | 91.3 | 86.2 | 43.9 | 43.2 | 44.7 | **58.0** | 52.6 | 64.5 |

methods and the results are shown in Table 1. We can find that our method achieve state-of-the-art performance on both three datasets. Specifically, our method can outperform the existing SOTA methods, *i.e.*, CLIP-RC (Zhang et al., 2024) and OTSeg+ (Ye et al., 2024), by a large margin, *i.e.*, 2.3%, 3.3%, and 0.3% in hIoU for PASCAL VOC, COCO-Stuff, and PASCAL Context dataset. When we dive deeper into the details of these results, we can find that our results come from the better generalization of the unseen categories. For example, In COCO-Stuff, the uIoU of our method is 4.3% higher than the SOTA methods and the same improvements can be seen across three benchmarks. For existing methods, their performance comes from overfitting to the seen categories.

We also provide a comparison of the computational cost and efficiency of our method with previous methods as shown in Table 2. Compared with the two-stage methods (first and second row in the table), our method can achieve a much higher inference speed and much lower GFLOPS. Compared with the methods that only add few trainable parameters, Though our trainable parameters are

Table 2: Comparisons in the efficiency between our method and other methods.

| Method | Parameter ↓ | GFLOPS ↓ | FPS ↑ |
|---|---|---|---|
| Zsseg (Xu et al., 2022) | 61.1 M | 1916.7 | 4.2 |
| ZegFormer (Ding et al., 2022a) | 60.3 M | 1829.3 | 6.8 |
| ZegCLIP (Zhou et al., 2023) | **13.8 M** | 61.1 | 25.6 |
| OTSeg+ (Ye et al., 2024) | **13.8 M** | 61.9 | 22.5 |
| Ours+SegNeXt (Guo et al., 2022) | 32.0 M | **33.5** | **40.9** |
| Ours+SETR (Zheng et al., 2021) | 91.0 M | 109.0 | 20.8 |
| Ours+Segformer (Xie et al., 2021) | 65.7 M | 60.7 | 23.0 |

higher than theirs, our method have high flexibility based on the segmentation model. For example, when we choose SegNeXt, an efficient segmentation model, our GFLOPS are nearly 50% of the SOTA one-stage methods and our inference speed is much faster.

## 4.2 ABLATION STUDIES

To evaluate the merits of the proposed methods, we conduct ablation studies. These experiments are conducted in the COCO-Stuff with 40K iterations. We use Segformer-B4 as backbones with all the hyperparameters unchanged.

**Ablation studies on the proposed methods.** We first ablate the proposed methods as shown in the first row of Table 3. We set the model without C2S distillation and latent prototypes as the baseline. As can be seen in the table, though its sIoU achieves 41.3%, its uIoU is very low with only 6.4%, leading to only 11.2% hIoU. By adding the C2S distillation, with the similar sIoU, the uIoU grows over 30% to 36.6%, resulting in a hIoU of 38.8%, indicating the effectiveness of C2S alignment. Finally, we add the latent embeddings with the pseudo masks to our method, we can find that the hIoU grows to 42.3% attributing to the large increase of uIoU which grows from 36.6% to 42.7%.

**Ablation studies on different distillations.** We use contrastive learning to distill the knowledge from CLIP in C2S distillation, here, we try to use different distillation methods to prove the effectiveness of our method as shown in Table 4. First, we change the contrastive distillation to the cosine similarity and find that though the sIoU achieves similar performance, the uIoU drops to

Table 3: Ablations on proposed modules.

| Methods | hIoU | sIoU | uIoU |
|---|---|---|---|
| baseline | 11.2 | 41.3 | 6.4 |
| baseline + distillation | 38.8 | 41.2 | 36.6 |
| baseline + distillation + latent embedding | **42.3** | **41.9** | **42.7** |

Table 4: Ablations on different distillations.

| Distillation | hIoU | sIoU | uIoU |
|---|---|---|---|
| Cosine Similarity (Tung & Mori, 2019) | 37.6 | 41.4 | 34.4 |
| L2 Loss (Wang et al., 2020) | 17.8 | 18.4 | 17.3 |
| Froster (Huang et al., 2024) | 37.2 | 41.7 | 33.6 |
| Our distillation | **42.3** | **41.9** | **42.7** |

Table 5: Ablations on global and local CLS tokens in latent embedding generation.

| Calibration | | hIoU | sIoU | uIoU |
|---|---|---|---|---|
| global | local | | | |
| - | - | 38.8 | 41.2 | 36.6 |
| - | ✓ | 41.9 | 41.3 | 42.6 |
| ✓ | - | 41.7 | 41.4 | 42.1 |
| ✓ | ✓ | **42.3** | **41.9** | **42.7** |

Table 6: Ablation on the input of latent embedding generation and prototype calibrator.

| Feature | Calibrator | hIoU | sIoU | uIoU |
|---|---|---|---|---|
| Prototypes | - | 41.0 | 41.5 | 40.5 |
| Prototypes | MLP | 41.5 | 41.3 | 41.8 |
| CLS tokens | - | 40.2 | 41.5 | 38.9 |
| CLS tokens | MLP | 40.9 | 41.5 | 40.3 |
| Prototypes + CLS tokens | Transformer | **42.3** | **41.9** | **42.7** |

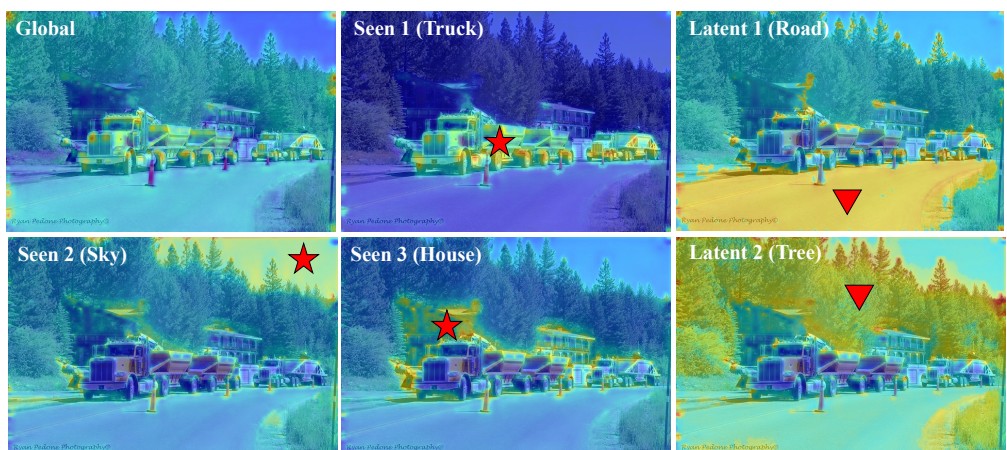

Figure 4: The similarities between the CLS tokens and the dense features.

34.4%. Then we change the cosine similarity to the direct L2 loss between the CLS tokens and the prototypes and find that both sIoU and uIoU drop drastically. Finally, we apply the residual feature distillation proposed in (Huang et al., 2024) and find that though a similar sIoU can be achieved, its uIoU is 9.1% lower than our method.

**Ablation studies on the latent embedding generation.** In this experiment, we want to clarify the effectiveness of the CLS tokens in the latent embedding generation as shown in Table 5. First, we set the methods without latent embedding as the baseline. Then we use only local CLS tokens to calibrate the latent embeddings and find that the hIoU improves due to the 6.0% improvements in uIoU. Then, we only use the global CLS tokens, we find that compared with local CLS tokens, the hIoU drops 0.2% due to the performance decreases in uIoU.

Besides, we also conduct experiments on how to calibrate the prototypes as shown in Table 6. First, we use the local prototypes $\mathbf{F}_u$ directly as the prototypes without any calibrator. Compared with our method, we find that the performance drops due to the uIoU. Then, we use MLP as the calibrator and find that compared with using only $\mathbf{F}_u$ the uIoU increases but is still lower than our method due to the lower IoU for unseen classes. Next, we directly apply the local CLS token as the prototype and find that the uIoU drops drastically to 40.2% from 42.3%. Finally, we add the MLP to the local CLS tokens and find the performance improvements.

### 4.3 QUALITATIVE ANALYSIS

**The visualization of the similarity between CLS tokens and dense features.** We want to find if the distillation can find the representative areas. Therefore, we visualize the similarities between the CLS tokens and the dense features as shown in Fig. 4. First, we visualize the similarities between

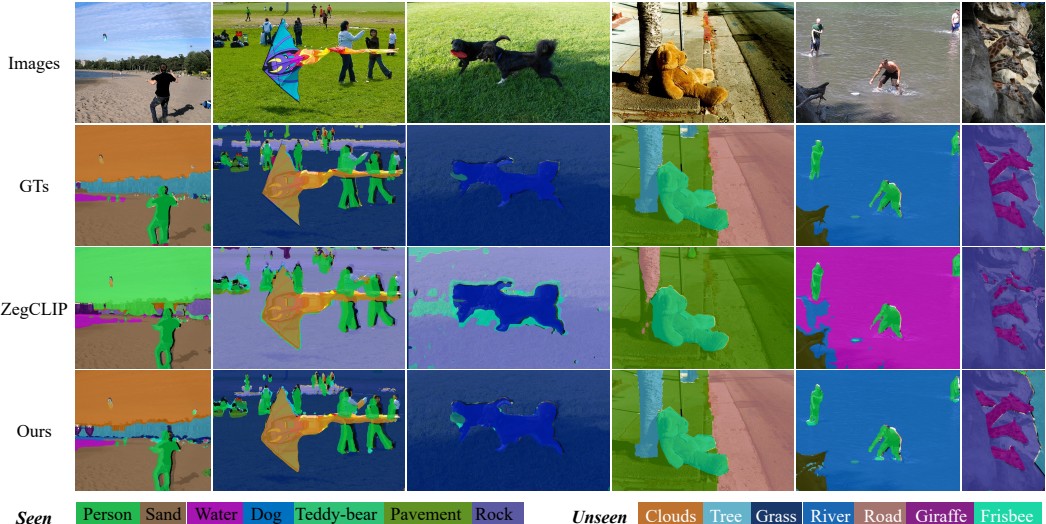

Seen: Person Sand Water Dog Teddy-bear Pavement Rock    Unseen: Clouds Tree Grass River Road Giraffe Frisbee

Figure 6: The similarities between the CLS tokens, including both global and local ones, and the dense features are illustrated. In this figure, the red stars represent areas corresponding to seen class labels, while the red triangles denote areas associated with latent (unseen) classes.

the global CLS tokens and the dense features. We can find that all the areas correspond to the global tokens. Then, we obtain local CLS tokens for the seen areas, *e.g.*, truck and house, and we can find that the correspondences are also class-specific. Finally, we generate pseudo maks for the unannotated areas, *i.e.*, road, and tree, and calculate their correspondence. We can also achieve the expected results.

**The visualization of the loss curves.** Fig. 5 shows the loss curves during training, with the overall loss (blue), global distillation loss (green), and local distillation loss (red) plotted over the number of iterations. Both global and local distillation losses decrease rapidly in the early stages and stabilize at lower values, indicating that the model efficiently learns from these distillation processes. The overall loss decreases more gradually but eventually stabilizes, reflecting the convergence of the model as training progresses.

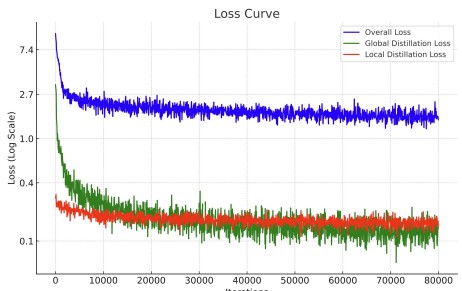

Figure 5: The loss curves during training.

**The visualization of prediction.** We visualize the prediction of our method as shown in Fig. 6. Compared with SOTA methods, *i.e.*, ZegCLIP (Zhou et al., 2023), our method can obtain exceptional results on both seen and unseen categories. For example, the 'trees' in the fourth image are classified as another unseen class (road) in ZegCLIP. However, our method can correctly recognize it. More visualizations can be seen in the ***Supplementary Materials***.

## 5 CONCLUSION

In this paper, we propose the CLIP-to-Seg Distillation framework to overcome the limitations of directly adapting CLIP for segmentation tasks. Our approach integrates both global and local distillation strategies to transfer CLIP's zero-shot generalization capabilities to closed-set segmentation models. By aligning feature prototypes from segmentation models with CLS tokens from CLIP at both global and local levels, we facilitate effective distillation from CLIP to pixel-level segmentation models. Additionally, introducing synthesized embeddings for latent classes enhances the model's ability to generalize to new concepts. Without adding extra parameters or computational overhead, our method achieves state-of-the-art performance on zero-shot segmentation benchmarks, offering a flexible and efficient solution to extend the generalization capabilities of existing segmentation models.

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

## A    APPENDIX

You may include other additional sections here.