# SUPPLIMEATARY MATERIALS FOR CLIP-TO-SEG DISTILLATION FOR INDUCTIVE ZERO-SHOT SEMANTIC SEGMENTATION

## 1 OTHER SETTINGS IN THIS PAPER.

**The name of unseen categories.** The names of the unseen categories in each dataset can be seen in Table. 1.

**Full experiment settings.** The proposed methods are implemented on the MMsegmentation (Contributors, 2020). The CLIP model applied in our method is based on the ViT-B/16 model and the channel ($C$) of the output text features is 512. All the experiments are conducted on 8 V100 GPUs and the batch size ($B$) is set to 16 for all three datasets. For all these three datasets, the size of the input images is set as 512 ($H$) $\times$ 512 ($W$). The iterations are set to 20k, 40k, and 80k for PASCAL VOC, PASCAL Context, and COCO-Stuff respectively. The optimizer is set to AdamW with the default training schedule in the MMSeg toolbox. In addition, the size of CLS tokens banks is set as 24, the threshold for mask merging $\lambda$ is 0.8, the size of the window in multi-scale K-Means is set as 3 and 7. $\tau$ in global loss is 0.07, and $\gamma$ is 1.0 for COCO-Stuff and 0 for PASCAL VOC and Context. $\alpha$ is set as learnable and $\beta$ is set as 2.

## 2 MORE DETAILS FOR MASK MERGING.

Formally, given the updated seed set $\mathbf{S}_{new}^{N_s \times C}$ where $N_s$ indicates the number of updated seeds, the corresponding mask $\mathbf{M}^{N_s \times H \times W}$ for each seed in $\mathbf{S}_{new}$, and the similarity threshold $\lambda$, we first calculate the cosine similarity $\mathbf{Simi}^{N_s \times N_s - 1}$ between each element in $\mathbf{S}'$ and all other elements. Then we find the maximum value $s_{\max}$ in $\mathbf{Simi}$. We find the row index $i$ of $s_{\max}$ and all the other values $\tilde{s}$ which are larger than $\lambda$ in the $i$th row. Then we add the masks belonging to $s_{\max}$ and $\tilde{s}$, and the merged masks serve as the pseudo label for one unknown category. And the $i$th seeds and the seeds belonging to $\tilde{s}$ can not be used again. We quit this loop until $s_{\max}$ is lower than $\lambda$. Finally, we concatenate the seen labels and the generated labels $\mathbf{Y}_g$ as fused labels $\mathbf{Y}_f$. The mask merging algorithm iteratively fuses regions that likely belong to the same category, making the dense features contain more coherent semantics. In addition, we show the pseudo-code of the proposed mask merging algorithm as shown in Algorithm. 1.

Table 1: Name of unseen categories.

| Dataset | Unseen Categories |
|---|---|
| VOC Everingham et al. (2015) | pottedplant, sheep, sofa, train, tvmonitor |
| COCO-Stuff Caesar et al. (2018) | cow, giraffe, suitcase, frisbee, skateboard carrot, scissors, cardboard, clouds, grass playingfield, river, road, tree, wall concrete |
| Context Mottaghi et al. (2014) | cow, motorbike, sofa, cat, boat, fence, bird, tv monitor, keyboard, aeroplane |

## 3 QUALITATIVE RESULTS

**The roles of latent class mining.** We further present additional results on the pseudo labels generated by the latent class mining algorithms, as shown in Fig. 1 and Fig. 2. These figures highlight the capability of our approach to discover latent classes, even when operating in small batches. Specifically, the results depicted are obtained from a batch of only two images, yet our method is

---

**Algorithm 1** Mask Merging Algorithm in Pytorch sytle

---

1: Initialize the threshold $\lambda$, and the masks $\mathcal{M}^{N_c * H * W} \in [0, 1]$ clustered by $\mathbf{S}_d^{N_c \times C}$ # Initialize the threshold for merging and the masks to be fused.
2: Initialize $\mathbf{Y}_{latent} = \varnothing$ # Initialize the merging results.
3: $\mathbf{S}_s = \cos(\mathbf{S}_d, \mathbf{S}_d)$
4: $s_{max} = \max(\mathbf{S}_s)$
5: **while** $s_{max} >= \lambda$ **do**
6:     $i = \text{where}(\mathbf{S}_s, \mathbf{s}_{max})[0]$ # Find the row where the max value is.
7:     $mask = \mathbf{S}_s[i] > \lambda$ # Find all the masks that have high similarity with the selected mask.
8:     $\mathbf{Y}_{latent}$.append($\mathcal{M}[mask]$.sum(0))# Add the temporal results to $F$.
9:     $\mathbf{S}_s[mask,:] = -\infty$ # The masks selected can not be selected again.
10:     $\mathbf{S}_s[:,mask] = -\infty$ # The masks selected can not be selected again.
11:     $s_{max} = \max(\mathbf{S}_s)$
12: **end while**
13: **return** $\mathbf{Y}_{latent}$

---

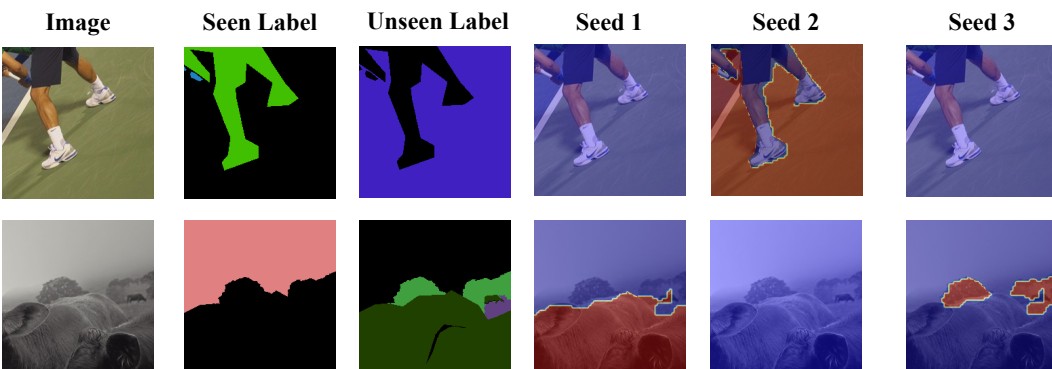

Figure 1: The pseudo labels generated by latent classes mining. In seed1, the category 'cows' can be found. In seed2, the unseen category 'playing field' can be found. In seed3, the unseen trees can be found.

robust enough to detect consistent latent classes across different batches. Moreover, our approach can identify and segment novel' objects that are not annotated in the original dataset, demonstrating its potential for discovering unseen or unannotated entities. For instance, in Fig. 2, the blue indicator in the center of the image is successfully detected by our model despite being unannotated in the ground truth labels. This example underscores the versatility and effectiveness of our method in recognizing latent classes and uncovering hidden object categories within complex scenes, extending beyond the scope of the provided annotations.

**The visualization of prediction.** Fig. 3 presents a visual comparison between the ground truth (GT), ZegCLIP predictions, and our proposed method for zero-shot semantic segmentation across both seen and unseen classes. The first row shows the input images, which include various objects such as cows, buses, and giraffes in different environments. The second row displays the GT, representing the manually labeled segmentation masks. The third row illustrates the predictions from ZegCLIP, which, while effective in some cases, exhibits inaccuracies in capturing object boundaries and details, particularly with unseen classes. The final row showcases the results from our method, which demonstrates improved segmentation accuracy and better alignment with the GT, especially in terms of object boundaries and overall segmentation quality, both for seen and unseen categories.

## REFERENCES

Holger Caesar, Jasper Uijlings, and Vittorio Ferrari. Coco-stuff: Thing and stuff classes in context. In *CVPR*, 2018.

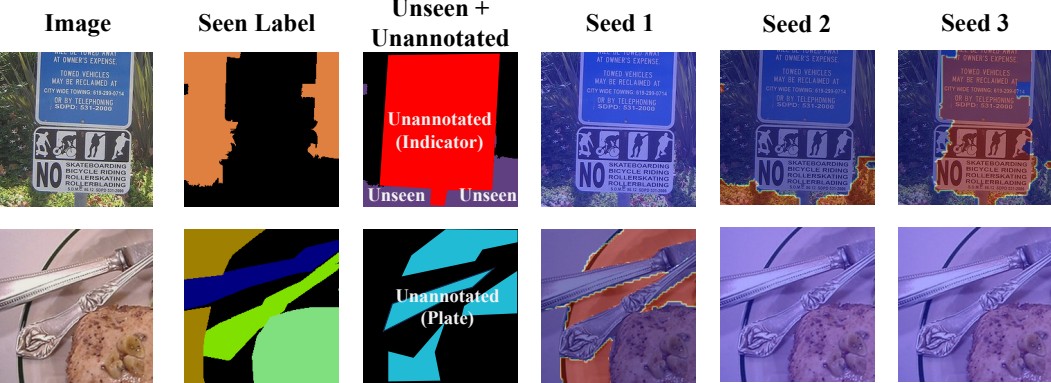

Figure 2: The pseudo labels generated by latent classes mining. In seed2, the unseen categories can be found. In seed1, the unannotated category 'plate' and the 'indicator' in seed3 can be found. Moreover, these three categories are separated.

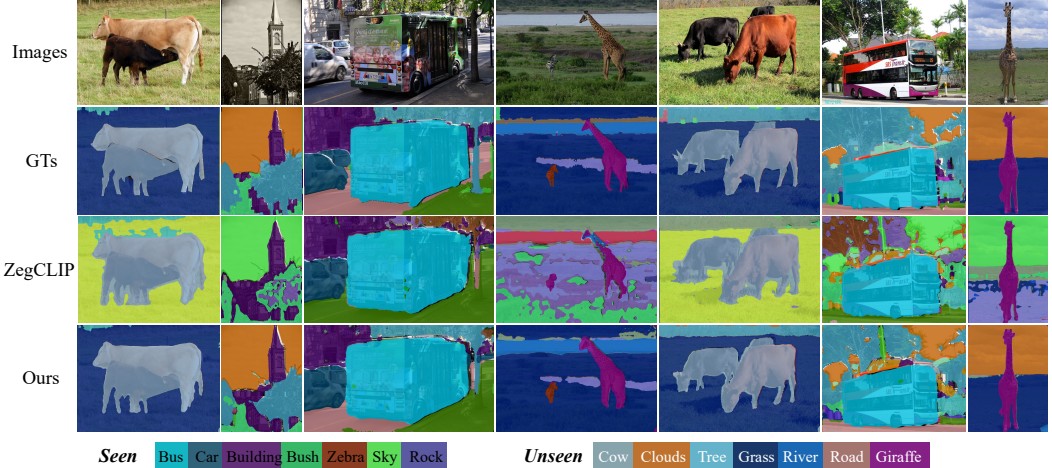

Figure 3: More visualization of the prediction.

MMSegmentation Contributors. Mmsegmentation: Openmmlab semantic segmentation toolbox and benchmark, 2020.

M. Everingham, S. M. A. Eslami, L. Van Gool, C. K. I. Williams, J. Winn, and A. Zisserman. The pascal visual object classes challenge: A retrospective. *International Journal of Computer Vision*, 2015.

Roozbeh Mottaghi, Xianjie Chen, Xiaobai Liu, Nam-Gyu Cho, Seong-Whan Lee, Sanja Fidler, Raquel Urtasun, and Alan Yuille. The role of context for object detection and semantic segmentation in the wild. In *CVPR*, 2014.