# OpenReview forum: "CLIP-to-Seg Distillation for Inductive Zero-shot Semantic Segmentation"
_ICLR.cc/2025/Conference — ICLR 2025 Conference Withdrawn Submission_

### Official Review · Reviewer_pQsq · 2024-10-30

**Soundness:** 1
**Presentation:** 3
**Contribution:** 2
**Rating:** 5
**Confidence:** 4

**Summary:**

This paper focuses on zero-shot semantic segmentation tasks, leveraging the robust zero-shot generalization capability of the CLIP model to enhance conventional segmentation models. The core contribution of this paper is a novel CLIP-to-Seg distillation approach, which adapts the CLIP model for segmentation by integrating global and local distillation collaboratively. Experiments show that this method achieves significant improvements.

**Strengths:**

1. The motivation and objective of this paper are meaningful, aiming to bridge the gap between the CLIP model and segmentation models.

2. The methods stay in efficient and low cost.

3. The method is model-independent and applicable to a wide range of down-stream segmentation models.

4. The proposed combination of global and local distillation is intriguing and appears to facilitate fine-grained tasks, such as segmentation.

**Weaknesses:**

1. One of the main supporting points is that conventional knowledge distillation requires feature size matching between teacher and student models. However, this is not a significant issue, as various methods—such as feature resizing, adaptive modules, and feature alignment—have been widely used in previous studies and papers.

2. The most important concern is as follows; please correct me if I am mistaken. I notice that this paper employs different backbones compared to previous work, which may render the comparison unfair, as it is unclear whether the observed improvements are due to the proposed method or the more powerful backbone. Furthermore, the backbone significantly influences performance in semantic segmentation tasks.

3. The paper has already presented the efficiency comparison. I am also curious about the time consumption, particularly since some operations appear to be time-intensive.

4. This method introduces several hyperparameters, such as window size and threshold. What is the influence of these hyperparameters, and will they increase the difficulty of training the model?

**Questions:**

1. Suggestion: The use of standard symbols for specific representations in the equation may create ambiguity and lead to potential confusion.

2. Please refer to the weaknesses part.

---

### Official Review · Reviewer_H9uH · 2024-11-02

**Soundness:** 3
**Presentation:** 2
**Contribution:** 2
**Rating:** 5
**Confidence:** 4

**Summary:**

This paper proposes a CLIP-to-Seg Distillation framework to overcome some drawbacks of directly adapting CLIP for image segmentation. By utilizing local-to-global distillation, the proposed approach can achieve state-of-the-art performance on multiple zero-shot segmentation benchmarks. The experiments are solid and can confirm the effectiveness of the proposed framework. However, the technical presentation is somewhat confusing, and the innovation of the approach is not clearly articulated.

**Strengths:**

The proposed approach can achieve state-of-the-art performance on zero-shot segmentation benchmarks, such as PASCAL VOC, COCO-Stuff， and PASCAL Context, which demonstrates the effectiveness of the CLIP-to-Seg Distillation framework.

**Weaknesses:**

(1)	Some recent works have studied the adaption of CLIP to zero-shot segmentation by knowledge distillation. What is the innovation of the proposed framework?
(2)	How can we ensure the accuracy of latent class generation, particularly in areas where similar semantics are present but correspond to different objects?
(3)	What is the relation between the latent embedding generation and local-to-global distillation?
(4)	How to capture the spatial information, considering that local distillation only accounts for the feature consistency of individual objects?

**Questions:**

See weakness.

---

### Official Review · Reviewer_Gmsn · 2024-11-03

**Soundness:** 3
**Presentation:** 3
**Contribution:** 2
**Rating:** 5
**Confidence:** 3

**Summary:**

This paper proposes a CLIP-to-Seg Distillation approach for inductive zero-shot semantic segmentation, aimed at transferring CLIP's strong zero-shot generalization abilities to traditional segmentation models. The method includes both global and local distillation to effectively transfer CLIP’s global and pixel-level semantic understanding, while reducing computational overhead by avoiding CLIP dependence during inference. This approach achieves state-of-the-art performance across various benchmarks.

**Strengths:**

1. The proposed method achieves significantly higher inference speed and lower GFLOPS.
2. The distillation strategy demonstrates effectiveness through ablation experiments.

**Weaknesses:**

1. The method lacks originality; leveraging CLIP for zero-shot segmentation and using pseudo-mask generation for latent class mining are already common practices.
2. Global and local distillation are also widely adopted in this area.
3. Some formula expressions are confusing, such as Y = {Y_i}i = 0^O.
4. Comparison with the popular Segment Anything v2 model is missing.
5. The significance of the loss curve visualizations is unclear.

**Questions:**

Please refer to the Weakness part.

---

### Official Review · Reviewer_J115 · 2024-11-04

**Soundness:** 2
**Presentation:** 3
**Contribution:** 2
**Rating:** 5
**Confidence:** 3

**Summary:**

This paper proposes a CLIP-to-Seg Distillation method to improve zero-shot segmentation. Traditional CLIP adaptations for segmentation face issues with mismatched objectives and added computational cost. The authors address this with global and local distillation techniques: global distillation uses CLIP’s high-level concepts, while local distillation adapts object-level features to discover unseen classes. This approach enhances closed-set segmentation models, enabling them to generalize to open classes without extra inference overhead, achieving state-of-the-art results on zero-shot segmentation benchmarks.

**Strengths:**

1. Introduces CLIP-to-Seg Distillation, an innovative approach combining global and local distillation with latent embeddings to extend CLIP's zero-shot capabilities to segmentation.

2. Clear organization and effective visuals make complex ideas accessible, presenting challenges and solutions in a concise manner.

3. Advances zero-shot segmentation by enabling inference without CLIP, achieving strong generalization on unseen classes and surpassing benchmarks, thus offering practical value for broader segmentation tasks.

**Weaknesses:**

1. Pixel-wise classification-based segmentation is unlikely to be more efficient than mask classification-based segmentation. The computational complexity for pixel-wise classification is O(H × W × C × K), whereas for mask classification, it is O(N × C × K). Here,
N represents the number of masks, which is generally much smaller than H×W. When the number of classes K is large, the computational demands of pixel-wise classification increase significantly. I noticed the lack of experiments on ADE20k-full, which includes over 800 classes. Theoretically, pixel-wise classification becomes inefficient as the number of classes grows, especially when we anticipate over 3,000 classes in the future. The most efficient approach would involve eliminating the additional CLIP classification from the two-stage model and distilling it into the two-stage framework.
2. The idea of global and local views is very common in contrastive learning [1] [2]. The method that use local and global views for contrasting distillation is not very novel.
3. How do the authors get the binary masks for unseen classes (Y_u)? Does that violate the zero-shot segmentation setting?
4. There are some writing issues. For example, in Figure 3, it is not suitable to call the process as mask pooling in the token generation. It looks like a crop of the masked image. There is a typo in Figure 3: "distillation" is not correct.


[1] Unsupervised Learning of Visual Features by Contrasting Cluster Assignments
[2] Propagate Yourself: Exploring Pixel-Level Consistency for Unsupervised Visual Representation Learning

**Questions:**

See the weakness

---

### Note · Authors · 2024-11-13

I have read and agree with the venue's withdrawal policy on behalf of myself and my co-authors.